# Respiratory Movements at Different Ages

**DOI:** 10.3390/medicina59061024

**Published:** 2023-05-25

**Authors:** Zhen-Min Bai, Yi-Ting Sun, Wen-Ming Liang, Inga Truskauskaitė, Miao-E Yan, Chun-Ri Li, Jing Xiao, Maiwulamu Aihemaiti, Lei Yuan, Osvaldas Rukšėnas

**Affiliations:** 1School of Sports Medicine and Rehabilitation, Beijing Sport University, Beijing 100084, China; 2Life Sciences Center, Vilnius University, LT-01513 Vilnius, Lithuania; 3Institute of Psychology, Vilnius University, LT-01513 Vilnius, Lithuania; 4Department of Traditional Chinese Medicine, Maternal and Children’s Healthcare Hospital of Beijing Dongcheng District, Beijing 100007, China; 5College of Acupuncture and Moxibustion, Liaoning University of Traditional Chinese Medicine, Shenyang 110847, China; 6Faculty of Health, Slovak Medical University, 974 05 Banská Bystrica, Slovakia; 7Department of Physiotherapy and Rehabilitation, Xiyuan Hospital, Chinese Academy of Chinese Medical Sciences, Beijing 100091, China; 8Department of Cardiology, Peking University Third Hospital, NHC Key Laboratory of Cardiovascular Molecular Biology and Regulatory Peptides, Beijing 100191, China

**Keywords:** aging, respiration, abdominal movement, abdominal motion, thoracic motion, vital capacity

## Abstract

*Background and Objectives:* The current study aimed to better understand the changes in respiration that occur with aging in men and women to provide accurate recommendations for breathing exercises to improve health. *Materials and Methods:* A total of 610 healthy subjects, aged 20 to 59, participated in the study. They performed quiet breathing while wearing two respiration belts (Vernier, Beaverton, OR, USA) at the height of the navel and at the xiphoid process to record abdominal motion (AM) and thoracic motion (TM), respectively. Vital capacity, representing maximal inhalation movement, was measured using a spirometer (Xindonghuateng, Beijing, China). After exclusion, 565 subjects (164 men, aged 41 ± 11; 401 women, aged 42 ± 9) were included for statistical analysis using the Kruskal–Wallis U test and stepwise multiple linear regression. *Results:* Abdominal motion and its contribution to spontaneous breathing were significantly larger for older men, while the contribution of thoracic motion was smaller for older men. There was no significant difference in thoracic motion between the younger and older men. The differences in women’s respiratory movements among various ages were mild and negligible. The contribution of thoracic motion to spontaneous breathing in women was larger than in men for those of older ages (40–59 years), but not for those of younger ages (20–39 years). Additionally, men’s and women’s vital capacities were less in those of older ages, and the men’s were larger than the women’s. *Conclusions:* The findings indicate that men’s abdominal contribution to spontaneous breathing increased from 20 to 59 years of age due to increased abdominal motion. Women’s respiratory movements did not change much with aging. The maximal inhalation movement became smaller with aging for men and women. Healthcare professionals should focus on improving thoracic mobility when addressing health concerns about aging.

## 1. Introduction

Globally, the size of the aging population is increasing rapidly [1]. To combat the challenges associated with aging and limited healthcare resources, medical professionals have begun advocating for the concept of “exercise is medicine” and promoting exercise as an essential component of healthcare [2]. Abdominal breathing (diaphragmatic breathing), either as a standalone exercise or combined with other movements, is becoming increasingly popular for improving health.

Abdominal breathing is characterized by the movement of the diaphragm, which intentionally expands the abdominal wall [3]. It is required to keep the chest stable and regulate respiration via the rise and fall of the diaphragm during abdominal breathing [4]. In contrast, thoracic breathing involves the use of accessory inspiratory muscles, such as external intercostal muscles, scalenus, pectoralis, and sternocleidomastoid, which increase the diameter of the thoracic cavity and form a breathing pattern dominated by thoracic movement [5,6]. Abdominal breathing exercise has been found to be beneficial for both physical and mental health [7], and previous research has demonstrated that voluntary breathing exercises (abdominal breathing) increase the contribution of abdominal motion during involuntary breathing (spontaneous breathing) [8]. Therefore, should people prioritize abdominal motion during spontaneous breathing for better health? Individuals who exhibit improper breathing patterns, such as rapid breathing and/or inhalation initiated with lifting of the chest, are recommended to enhance abdominal motion [9]. However, it remains unclear whether healthy individuals, especially those in the middle-aged population, should alter their breathing to a more abdominal-dominant pattern to prevent age-related health deterioration. Furthermore, it is not yet known whether increasing the proportion of abdominal movement adversely affects our health. This question should be a cause for concern, as a large number of individuals practice abdominal breathing worldwide. It is certain that physical performance begins to decline at approximately 30 years of age [10,11]. Understanding how respiratory movements change with age can provide greater confidence in answering the aforementioned question and allow for more targeted exercise recommendations.

There is some evidence regarding the changes in respiratory movement with aging. One study found that the thoracic contribution to spontaneous breathing decreased with aging while the abdominal contribution increased, and the authors proposed that the increased abdominal contribution was compensation for the decreased thoracic motion for maintaining the level of tidal volume [12]. Nevertheless, several previous studies reported no systematic age effects on respiratory movements during quiet breathing [13,14,15,16,17]. Noteworthily, the sample sizes were relatively small in all previous studies, and the biggest sample comprised 120 healthy subjects, combining 2 genders [14]. Sample size substantially affects statistical results [18], and men and women are different physiologically. It is necessary to increase a sample’s size to verify the result and to analyze men’s and women’s data separately, according to the concept of precision medicine. In addition, only one study that evaluated respiratory movements in a standing position was found [14]; thus, data regarding the standing position are needed. Furthermore, body size, including the circumference of the waist and hips, might be confounding factors when analyzing respiratory movements and age, and it should be considered. Apart from respiratory movement during spontaneous breathing, the capacity for maximal chest expansion is also crucial for our health. A study has reported a significant correlation between vital capacity and maximum thoracic expansion [19].

Based on the points mentioned above, the present study was designed to increase the sample size, examine various genders and age groups, control the impact of body size, and assess spontaneous and maximal forceful breathing. Ultimately, the primary goals were to investigate the correlation between respiratory movements and aging and to answer whether we should attempt to enhance abdominal motion in spontaneous breathing through intentional abdominal breathing exercises for health improvement.

## 2. Materials and Methods

### 2.1. Trial Design and Participants

This was a cross-sectional study. A total of 610 healthy participants were enrolled in the study through convenience sampling from 6 communities in the Haidian District in Beijing, as shown in Figure 1. Exact values from the different ages can clearly present the changes seen in the different ages and can be referred to future studies. Considering the sample size and referred previous study [20], the participants were divided into 4 age groups of 10-year intervals (approximately a generation). The data were collected from June to December in 2021. The inclusion criteria were 20–59 years old, capable of comprehending and answering the interview questions, having completed the Physical Activity Readiness Questionnaire and having met all the requirements, and having provided a signed informed consent. The exclusion criteria were pregnant or lactating women, having a mental illness, having acute diseases or having suffered from acute diseases and not yet being recovered physically, having consumed caffeinated drinks in the 2 h before the experiment, and having a respiration rate that was less than the mean minus 2 times the standard deviation (SD) or greater than the mean plus 2 times the SD; no 10 consecutive stable respiration cycles were found.

The Research Ethics Committee of Beijing Sport University approved the study (approval number: 2021079H), and all participants were informed of the test’s risks before signing the informed consent form.

### 2.2. Respiratory Movements Testing and Data Processing

Respiration belts (Vernier, Beaverton, OR, USA) were utilized to test respiratory movements in this study. These belts were fabric, with a resistive stretch sensor embedded in them, and they could be used to measure respiration rate, breathing maneuvers, and respiratory waveforms. For the current investigation, two belts were employed: one was secured at the xiphoid process level and the other at the navel level to monitor chest and abdominal movements (Figure 2). The participants were instructed to sit quietly for 5 min before testing to calm down physically and mentally. To remove the impact of abdominal circumference, the subjects stood up. The straps were adjusted by the researcher until the belts’ lights turned green or red, according to the user’s directions. During the test, the participants watched a neutral video of slow-swimming fish in the sea on an 11-inch Xiaomi Pad (resolution: 2560 × 1600; Xiaomi, Beijing, China), which was placed 50–80 cm away in front of them. The goal was to divert their attention from breathing.

The test lasted two minutes, and the two belts were used simultaneously. Previous studies used various methods to choose the number of respiratory cycles for analysis—from three satisfactory readings to six minutes of breathing cycles [22,23,24]. We noticed that the respiration waves generally reached a stable state after 30 s from the start of the test. Therefore, we chose 10 consecutive stable respiration cycles with minimal motion artifacts and baseline wandering after the first 30 s of the test. The data were then imported into OriginPro 9.0 (OriginLab, Northampton, MA, USA) from the Vernier Graphical Analysis (Vernier, Beaverton, OR, USA) to extract the peaks and troughs of the breathing waves. Subsequently, we imported the peaks and troughs into Excel (Excel, Microsoft, Redmond, WA, USA) to calculate the abdominal motion (AM), thoracic motion (TM), and respiration rate (RR). We calculated the AM and TM separately, and their values were determined as the 10 averaged peak (P) forces minus 10 average trough (T) forces (motion = (P1 + P2 … + P10)/10 − (T1 + T2 … + T10)/10). The respiration rate was calculated as 60 s divided by the time taken for 1 respiration cycle, which was derived from the difference between the time of the eleventh peak and the time of the first peak divided by 10 (RR = 60/(P11 − P1)/10) (Figure 1). The signals were presented as force (unit = Newton) and sampled at a frequency of 10 Hz.

### 2.3. Vital Capacity Testing

For the vital capacity (VC) test, the participants stood upright and held the handle of the spirometer (Xindonghuateng Sports Equipment Co., Ltd., Beijing, China). They took a full inspiration and then exhaled slowly and maximally. The test was performed twice, and the higher value was recorded with an accuracy of 1 mL.

### 2.4. Statistical Analyses

The Shapiro–Wilk test was used to assess the data distribution. A Spearman Correlation test was used to detect the influence of body size on respiratory movement. To test the differences between the four age groups, a Kruskal–Wallis analysis was used since normal and skewed distributions were mixed throughout the data. For the two-by-two comparison of the groups two by two, the Mann–Whitney U test was used. The Bonferroni corrected *p*-values were estimated to assess the significance at *p* < 0.05, and the effect sizes were derived from the *z* values divided by the square root of the sample size [25]. In addition, multiple regression analysis was used to test the links between the respiratory movements and age, with adjustments for body size. The sample size estimation considered the 5 independent variables in the model (age, weight, height, waist circumference, and hip circumference), and the equation *n* ≥ 30 + 10 k [26] was used, which resulted in a sample of at least 80 participants. The final model was determined from the adjusted coefficient (R^2^) and the statistical significance. To determine the statistical quality of the model, the multicollinearity was verified by the variance inflation factor, as well as the homogeneity and normal distribution of the residuals by graphic visual analysis. The level of significance was set at *p* < 0.05. SPSS was used (IBM Corp., Armonk, NY, USA).

## 3. Results

### 3.1. Males’ Respiratory Movements for the Different Age Groups

The male participants were divided into four age groups of ten-year intervals. As shown in Table 1, the men’s body weight, BMI, waist circumference, and hip circumference were not significantly different between the 4 age groups (*p* > 0.05). However, the body heights of the 20–29 group were significantly higher than those of the 40–49 group (*p* < 0.05), and the waist–hip circumference ratio was significantly higher in the older age groups compared to the younger groups (*p* < 0.0001). The older age groups’ AMs and AM/(AM + TM)s were larger than those of younger age groups, whereas the differences in TMs were not significant, indicating that the increased AM/(AM + TM) mainly contributed to the increases in the AMs. Specifically, the 50–59 group’s AM/(AM + TM)s were larger than those of the 20–29 group (*z* = −3.58, *p* < 0.001, effect size = 0.42) and the 30–39 group (*z* = −3.26, *p* = 0.001, effect size = 0.35). Because the correlation between height and AM/(AM + TM) was significant (*rho* = −0.304, *p* < 0.001), we controlled the influence of height by normalizing the AM/(AM + TM) ((AM/(AM + TM) × height). The results from the normalized values were almost the same as those from the non-normalized values (the 20s vs. 50s, *z* = −3.40, *p* < 0.001, effect size = 0.40; 30s vs. 50s, *z* = −3.23, *p* < 0.001, effect size = 0.34), which means that the abdominal contribution in the older group (50s) was larger than that in the younger groups (20s and 30s), apart from the confounding factor of body size, which had a medium effect size. The differences between the TM/(AM + TM) values were opposite to those for AM/(AM + TM).

Vital capacity decreased gradually and significantly with age. The 20s group’s vital capacities was larger than the 40s group’s (*z* = −4.85, *p* < 0.001, effect size = 0.56) and 50s (*z* = −6.22, *p* < 0.001, effect size = 0.72). The 30s group’s vital capacities were larger than the 40s’ (*z* = −3.05, *p* = 0.002, effect size = 0.32) and the 50s group’s (*z* = −4.64, *p* < 0.001, effect size = 0.5). Using height to normalize vital capacity (VC/height × 100) is a common way to eliminate the influence of body size, and we found that vital capacity was significantly correlated with height (*rho* = 0.304, *p* < 0.001). Therefore, we tested the vital capacity adjusted body heights and found the same results. In addition, some past studies have used body weight to adjust vital capacity. Therefore, we also tested vital capacity adjusted body weights and found that the only difference in the results was that the vital capacities of those in their 20s was significantly larger than the vital capacities of those in their 30s.

Regarding the stepwise multiple regression analysis, the AM, TM, AM + TM, AM/(AM + TM), TM/(AM + TM), and vital capacity were the dependent variables tested in separate analyses, while the age, height, weight, waist circumference, and hip circumference were the independent variables included in every test. Additionally, to prevent multicollinearity with height, weight, waist circumference, and hip circumference, we included the age, BMI, and waist–hip ratio as the independent variables and tested them with all the dependent variables. Weight, BMI, waist circumference, hip circumference, and waist–hip ratio were not significantly linked with the respiratory parameters (*p* < 0.05) (except for the waist circumference and vital capacity), which were not presented. Table 2 shows the results that were consistent with those from group comparison tests in Table 1, where the men’s AM/(AM + TM) values increased with age, and the thoracic contribution decreased. Further, both age and height were significantly associated with AM/(AM + TM), and the model explained 17.1% of the variance. In addition, AM/(AM + TM) was predicted to increase by 0.4% each year. The association of age with TM/(AM + TM) was opposite to that of the abdominal motion. Age was significantly associated with AM, and the model explained 13.2% of the variance. AM was predicted to increase for 0.043 Newton each year. In contrast, the association of age with thoracic motion was not significant, which was also in line with the results from the group comparison tests. Regarding the vital capacity, as shown in Table 2, age, height, and waist circumference were significantly linked with vital capacity, and the model explained 33.3% of the variance. In addition, vital capacity was predicted to decrease by 36.9 mL each year.

### 3.2. Females’ Respiratory Movements for the Different Age Groups

The female participants were also divided into four age groups using ten-year intervals (Table 3). The women’s body heights, weights, BMIs, and hip circumferences were not significantly different between the 4 age groups (*p* > 0.05). Nevertheless, the waist circumference and waist–hip circumference ratio were larger in the older groups than in the younger groups (*p* < 0.05 and *p* < 0.01, respectively). However, the Spearman correlation test revealed no significant correlations between the women’s body sizes (heights, weights, BMIs, waist circumferences, hip circumferences, and waist–hip ratios) and the AM, TM, AM + TM, AM/(AM + TM), and TM/(AM + TM) values (*rho* < 0.2, *p* < 0.5). This suggested that body size did not act as a confounding factor when comparing spontaneous respiratory movements among the different age groups. Furthermore, comparison tests demonstrated no significant differences in the AM, TM, AM + TM, AM/(AM + TM), and TM/(AM + TM) values between the age groups (Table 3).

The vital capacities of the women in the younger age groups were larger than those of the older women (Table 3). The vital capacities of the women in their 20s were larger than those of the women in their 50s (*z* = −5.0, *p* < 0.001, effect size = 0.43). The vital capacities of the women in their 50s was smaller than those of the women in their 30s (*z* = −5.79, *p* < 0.001, effect size = 0.40) and 40s (*z* = −3.35, *p* < 0.001, effect size = 0.22). Additionally, as the women’s vital capacity was significantly correlated with their heights (*rho* = 0.26, *p* < 0.001), the normalized vital capacities to both height and weight were tested, and the results were comparable to the non-normalized values.

When testing the relationship between the women’s respiration movements and their ages, the results showed a quadratic U-shaped model, and the trough was located at the age of 40. Therefore, we split the age into 2 parts—from 20 to 39 years old and from 40 to 59 years old—for linear regression analysis. As shown in Table 4, although the AM/(AM + TM) and TM/(AM + TM) values were significantly associated with age, the model only explained 2.5% of the variance. The AM, TM, and AM + TM values were not presented, as they were not significantly predicted by age and body size. These findings indicated that women’s spontaneous respiratory movements were not substantially associated with age. In contrast, their vital capacities decreased significantly with age. Age and height were linked with the women’s vital capacities, and the model explained 17.4% of the variance. In addition, the vital capacity was predicted to decrease by 21.8 mL each year.

### 3.3. Comparisons of Respiratory Movements between Men and Women for the Different Age Groups

In the 20s and 30s groups, there were no significant differences observed between the men and women for the AM/(AM + TM) and TM/(AM + TM) values. However, in the age 40s and 50s groups, as indicated in Table 5, there were significant differences. The men had higher AM, TM, AM + TM, VC, and normalized VC values than the women for all age groups, except for the AM values in the 20s age group and the VC/weight values in the 30s age group.

## 4. Discussion

The current study aimed to investigate the changes in respiratory movements that occur with age in men and women to provide accurate recommendations for breathing exercises to improve health. Therefore, we recruited 610 participants, aged 20 to 59, to investigate the respiratory movements of the different ages and genders. The data from the men and women were analyzed separately, and each gender was divided into four age groups to specify the differences. Overall, we found that older men’s abdominal contributions to spontaneous breathing were significantly larger than those in the younger men (*p* < 0.001), whereas the changes in the women’s respiratory movements were unsubstantial (*p* > 0.05). In addition, the vital capacities (the ability to maximally expand the rib cage) decreased with age for both the men and women (*p* < 0.001).

### 4.1. Men’s Respiratory Movements for the Different Age Groups

Regarding the respiratory movements during spontaneous breathing, both comparisons and correlational analyses were used to verify and consolidate the results. We confirmed that there were more abdominal contributions and fewer thoracic contributions in the more advanced age groups compared to the younger groups for the men. We also confirmed that the abdominal contribution was positively correlated with age, while the thoracic contribution was negatively correlated with age for the male participants aged 20 to 59 years old. This finding was consistent with one past study [12], but it was inconsistent with other studies [13,14,15,16,17]. Due to the larger sample size of the present study, our results strengthen the existing findings. Regarding vital capacity/the ability to maximally expand the chest wall, it decreased gradually and significantly with age, which was in line with the results of other studies [17,27]. 

The chest wall compliance reduces in the process of aging [28], which can be caused by increased rigidity of the ribs [29], thoracic kyphosis [30], and reductions in respiratory muscle strength [31]. Furthermore, the rib end-to-end separations and rib aspect ratios are seen to increase with age, producing elongated and flatter overall rib shapes in elderly populations [32]. The barrel-shaped thoracic walls found in older populations reduces the mobility around the involved joints and constrains the activity of the ribs [33]. The changes in muscle units could also be an influential factor. The muscle units are classified into slow-twitch (type S) and fast-twitch (type F) motor units, and type F motor units are further sub-classified into fast-twitch fatigue-resistant (type FR), fast-twitch fatigue-intermediate (type FInt), and fast-twitch fatigable (type FF) motor units [34]. During eupneic ventilation (spontaneous breathing), only type S and type FR motor units in the diaphragm are recruited, whereas forced ventilation requires additional recruitment of type FInt and type FF motor units [35,36]. Sarcopenia is a common phenomenon in aging populations [37]. Type S motor units are the ones most saved by sarcopenic processes, while type F are the ones most negatively affected [34]. Therefore, these changes in the chest wall compliance and muscle units deteriorate vital capacity/maximal chest expansion ability, whereas the thoracic motion during spontaneous breathing is not significantly affected.

Concerning the increase in abdominal motion during spontaneous breathing in advancing ages, Mendes et al. (2020) proposed that the change in abdominal motion was a compensatory mechanism triggered by decreasing thoracic movements [12]. Andrea et al. (2012) found that the thickness of the diaphragm and its contractility are minimally affected by age, as diaphragm thickness in the zone of apposition remains stable throughout a wide age range (20–83), with a mean thickness of 3.3 mm, and diaphragm contractility also does not change significantly with age [38]. Moreover, Özden et al. (2019) found that the diaphragm was significantly thicker in the older group (age: 71.3 ± 5.2 years; thickness: 2.3 ± 0.6 mm) than in younger adults (age: 26.9 ± 5.1 years; thickness: 2.0 ± 0.5 mm), and they suggested that the thickening of the diaphragm could be attributed to substantial atrophy in the other core muscles in the older groups to preserve balance and posture [39]. Because the diaphragm is less influenced or even becomes thicker with age, its dome-like shape presses down on the abdominal cavity during inspiration, resulting in increased abdominal movement. In addition, Mendes et al. found for each year of increase in age (for mixed men and women between 21 and 85 years of age), the abdomen (navel level) percentage contribution was increased by 0.29%, the pulmonary rib cage (axilla level) percentage contribution was reduced by 0.20%, and the abdominal rib cage (xiphoid process level) percentage contribution reduced by 0.08% [12]. Our findings were consistent with the study by Mendes et al., and we agreed with them that the increased proportion of abdominal movement compensates for the limited movement of the chest. Differently, the present study stratified men and women and found that women’s respiratory movements did not change substantially with age, which will be discussed in the next paragraph.

### 4.2. Women’s Respiratory Movements for the Different Age Groups

The changes in women’s respiration movement were not substantial, although we observed a U-shaped change from 20 years old to 59 years old. Except for the finding from the study conducted by Mendes et al., wherein the abdominal contribution increased with age (for mixed men and women) [12], other studies have consistently observed nonsignificant (*p* > 0.05) changes in women’s abdominal contributions [13,14,15,16]. Generally, we confirmed the results from most of the previous studies.

Women have a greater inclination of the ribs and lower radial dimension of the rib cage than men, which is considered an adaptation and evolution of the reproductive system to accommodate a growing fetus during pregnancy [40]. During pregnancy, the augmented tidal volume necessary to meet reproductive needs is mainly attained through an enhanced displacement of the ribcage without any consistent changes being detected in the abdominal contribution, as measured by magnetometers [41]. Therefore, the women exhibited prominent thoracic movements during quiet breathing. Due to the anatomical characteristics of women, they are more likely to use chest breathing. The degree of restriction of the thorax during the aging process for women should be lighter than that for men. As a result, the compensatory increase in diaphragmatic movement is less, and there is not much change in abdominal motion.

There was a mild U-shaped change in women’s respiratory movements with decreased abdominal contribution from 20 years of age to 39 years of age, and increased from 40 years of age to 59 years of age. Data variations can cause these results, but this phenomenon might also be influenced by sociopsychological factors, such as stress. A past survey (where the participants came from the region where the present study was conducted) revealed that younger people (20–39 years of age) have significantly higher occupational stress than older people (40–59 years of age), and women have significantly higher levels of occupational stress than men [42]. When experiencing occupational stress, people may hyperventilate, causing biomechanical stress in their neck and shoulder region due to the activation of the sternocleidomastoid, scalene, and trapezius muscles, which tend to increase thoracic motion [43]. We assumed that when stress is reduced after 40 years of age, the activation of supplementary respiratory muscles (e.g., the sternocleidomastoid, scalene, and trapezius muscles) decreases, and the contribution of abdominal motion increases.

Regarding the changes in the women’s vital capacities, the results were comparable to those of the men, showing decreases with age. The anatomical and physiological changes responsible for these decreases are likely the same as those discussed previously for men.

### 4.3. Comparison between Women and Men

The abdominal and thoracic contribution to spontaneous breathing did not significantly differ (*p* > 0.05) between the men and women aged 20–29 and 30–39, but they showed significant differences for those aged 40–49 and 50–59. The men had larger abdominal contributions to spontaneous breathing in the older age groups, with an increasing trend, while the women’s abdominal contributions remained relatively stable. As a result, no significant differences in the abdominal contributions existed between the men and women aged 20–39, but differences emerged in those aged 40–59. No rational explanations for these changes exist, except for those previously mentioned. Regarding abdominal motion, thoracic motion, and the sum of abdominal and thoracic motions, the men demonstrated higher values, which should be reasonable, as they were taller and heavier than the women in the present study.

Regarding the normalized and absolute vital capacities, the men’s capacities were greater than women’s for every age group, which was consistent with a previous study [44]. Various techniques, including standard morphometric methods, chest radiographs, and three-dimensional geometric morphometric methods on computed tomography scans, have demonstrated that men possess larger lungs compared to women [45]. Women have smaller rib cages and lower respiratory muscle strength compared to men [40,46]. These reasons are supposed to be the main causes for women having smaller vital capacity.

### 4.4. Practical Suggestions

The long-term practice of abdominal breathing, or diaphragmatic breathing, which involves expanding the abdomen and restricting the chest wall, may not be advisable for men, particularly those who are elderly, because this breathing technique involves very little movement of the chest, which can lead to stiffness in the chest [6]. We recommend retaining the natural expansion of the chest during abdominal breathing exercises. To improve the ability of chest expansion, dirga pranayama (the three-part breath) should be considered, as it requires the full expansion of both the abdomen and the rib cage [47].

### 4.5. Limitations

The current study had at least three limitations. Firstly, the sample size across the four age groups was unbalanced, with female participants outnumbering males and more elderly participants compared to young participants. Secondly, the study did not collect information about the participants’ smoking histories, which may have been an influential factor in participants’ respiratory movements. Thirdly, the data were collected from June to December 2021 during the COVID-19 outbreak. The World Health Organization reported that shortness of breath or difficulty breathing are common symptoms of post-COVID-19 conditions [48]. Unfortunately, we did not record whether the participants had ever had a coronavirus, but we excluded people who continued to have cardiopulmonary diseases.

## 5. Conclusions

This study found significant differences in the thoracic and abdominal movements during spontaneous breathing among men of different ages. Specifically, abdominal contributions were larger in older men than in younger men, and thoracic contributions were smaller in older men than in younger men. The women’s respiratory movements were not significantly different at the various ages. Between genders, women aged 40 to 59 had a lower abdominal contribution and larger thoracic contribution than the men of the same age range. Vital capacities decreased with age for both the men and women. For healthy individuals, it is not recommended to attempt to increase abdominal motion through breathing exercises. Instead, healthcare professionals should focus on improving thoracic mobility when addressing health concerns related to aging.

## Figures and Tables

**Figure 1 medicina-59-01024-f001:**
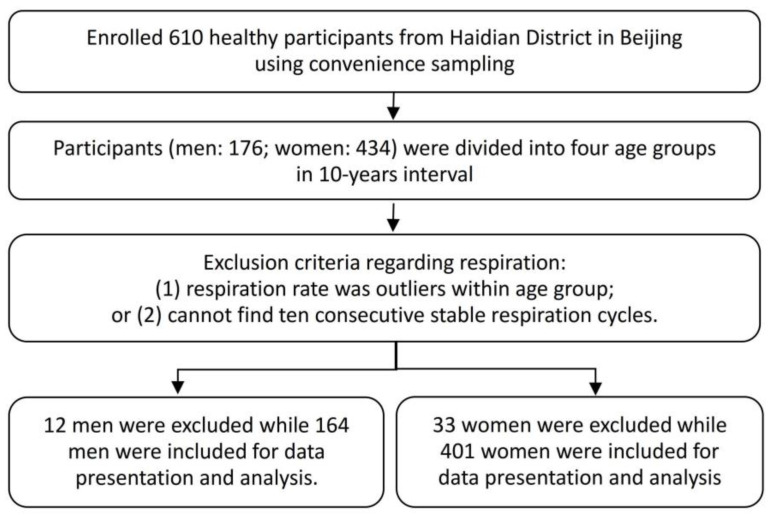
Flowchart of the sampling and grouping methods.

**Figure 2 medicina-59-01024-f002:**
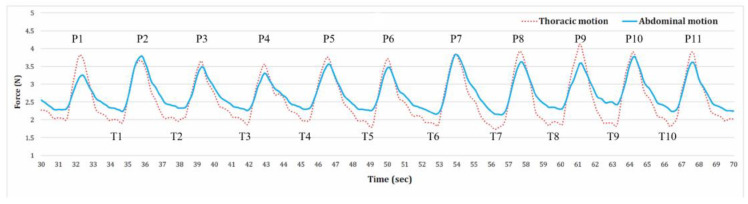
Wave lines of respiration recorded from two respiration belts. The solid line represents the abdominal motion (at the naval level), and the dotted line represents the thoracic motion (at the xiphoid process level). P = peak; T = trough. This figure was adapted from our previous publication [21].

**Table 1 medicina-59-01024-t001:** All tested values and the comparisons between the differently aged men.

Group (Age)	1 (20–29) N = 28	2 (30–39) N = 42	3 (40–49) N = 47	4 (50–59) N = 46	z	*p*	Between-Group Comparison
Age	25 (5)	35 (4)	43(6)	54 (5)	151	<0.001	All groups were different
Height (cm)	177 (6)	175 (10)	173 (8)	172 (10)	9.84	0.020	1 vs. 3 *
Weight (kg)	75.0 (20.9)	82.2 (15.9)	73.3 (18.2)	74.1 (10.4)	5.79	0.122	
BMI (kg/m^2^)	25.1 (6.44)	25.9 (4.79)	25.4 (5.43)	25.2 (4.06)	2.92	0.404	
Waist Circumference (cm)	87.0 (18.8)	90.3 (12.0)	91.4 (15.3)	92.4 (9.50)	6.23	0.101	
Hip Circumference (cm)	100 (8.78)	103 (10.3)	97.8 (10.5)	98.1 (6.40)	5.66	0.130	
Waist–hip ratio	0.86 (0.13)	0.89 (0.09)	0.92 (0.08)	0.94 (0.06)	25.0	<0.001	1 vs. 3 *, 1 vs. 4 **, and 2 vs. 4 **
Respiration rate (rep/min)	14.7 (4.04)	16.9 (5.32)	15.7 (5.25)	18.1 (5.54)	8.64	0.034	1 vs. 4 *
AM (N)	1.57 (1.70)	1.28 (1.21)	2.02 (1.80)	2.37 (1.83)	21.7	0.000	1 vs. 4 **, 2 vs. 3 *, and 2 vs. 4 **
TM (N)	2.85 (1.55)	3.21 (1.70)	3.34 (2.41)	3.46 (2.31)	1.03	0.793	
AM + TM (N)	4.39 (3.10)	4.32 (2.02)	5.88 (4.01)	5.96 (2.34)	10.2	0.017	
AM/(AM + TM) (%)	28.3 (23.1)	31.1 (26.1)	37.1 (21.8)	42.3 (20.7)	18.0	<0.001	1 vs. 4 **,2 vs. 4 **,
TM/(AM + TM) (%)	71.7 (23.1)	68.9 (26.1)	62.9 (21.8)	57.7 (20.7)	18.0	<0.001	1 vs. 4 **,2 vs. 4 **
Vital capacity (ml)	4169 (848)	3833 (1207)	3335 (944)	3120 (677)	49.4	<0.001	1 vs. 3 **,1 vs. 4 **, 2 vs. 3 *, and 2 vs. 4 **
Vital capacity/height (mL)	23.5 (4.44)	22.5 (6.66)	19.3 (5.58)	18.1 (3.70)	46.1	<0.001	1 vs. 3 **,1 vs. 4 **, 2 vs. 3 *, and 2 vs. 4 **
Vital capacity/weight (mL)	51.2 (16.1)	48.4 (18.1)	45.4 (20.9)	41.2 (12.8)	20.6	<0.001	1 vs. 3 *, 1 vs. 4 **, and 2 vs. 4 *

Note: all values are presented as medians (interquartile ranges). Under “Between-group comparison”, 1 represents the 20–29 group, 2 represents the 30–39 group, 3 represents the 40–49 group, and 4 represents the 50–59 group. * *p* < 0.05, ** *p* < 0.01; AM, abdominal motion; TM, thoracic motion; AM + TM, the sum of the abdominal motion and the thoracic motion; AM/(AM + TM), the abdominal motion divided by the sum of the abdominal and thoracic motions; BMI, the weight in kilograms divided by the square of the height in meters; N, Newtons; kg, kilograms; cm, centimeters; ml, milliliters. The ages shown are in years.

**Table 2 medicina-59-01024-t002:** Regression analysis for the prediction of respiratory movement in men.

Dependent Variables	Independent Variables	B	95% CI	β	*p* ^β^	R^2^	F	*p* ^F^
AM	Age	0.043	0.026, 0.061	0.363	<0.001	0.132	24.345	<0.001
AM/(AM + TM)	Age	0.004	0.002, 0.006	0.265	<0.001	0.171	16.355	<0.001
	Height	−0.007	−0.010, −0.003	−0.264	<0.001			
TM/(AM + TM)	Age	−0.004	−0.006, −0.002	−0.265	<0.001	0.171	16.355	<0.001
	Height	0.007	0.003, 0.010	0.264	<0.001			
Vital capacity	Age	−36.941	−47.067, −26.815	−0.479	<0.001	0.333	27.816	<0.001
	Height	29.111	11.411, 46.820	0.216	<0.001			
	Waist circumference	−10.456	−20.429, −0.482	−0.135	0.040			

Note: as the stepwise regression analysis was performed, the non-significant predictors were removed automatically. AM, abdominal motion; TM, thoracic motion; AM + TM, sum of the abdominal motion and thoracic motion; AM/(AM + TM), abdominal motion divided by the sum of the abdominal and thoracic motions; VC, vital capacity.

**Table 3 medicina-59-01024-t003:** All tested values and the comparison between differently aged women.

Group (Age)	1 (20–29) N = 40	2 (30–39) N = 114	3 (40–49) N = 146	4 (50–59) N = 101	z	*p*	Between-Group Comparisons
Age	26 (4)	36 (5)	44 (5)	55 (5)	365	<0.001	All groups were different
Height (cm)	162 (7)	161 (8)	162 (7)	160 (7)	3.57	0.312	
Weight (kg)	56.8 (9.80)	58.7 (14.6)	61.0 (11.9)	61.3 (11.5)	4.46	0.216	
BMI (kg/m^2^)	21.6 (4.45)	22.5 (5.70)	22.5 (4.07)	23.7 (4.21)	6.47	0.091	
Waist Circumference (cm)	71.0 (8.90)	75.3 (17.1)	76.2 (11.0)	79.9 (12.6)	19.5	<0.001	1 vs. 2 *, 1 vs. 3 *, and 1 vs. 4 **,
Hip Circumference (cm)	93.3 (5.50)	94.0 (11.8)	94.2 (8.60)	94.9 (9.20)	0.40	0.939	
Waist–hip ratio	0.75 (0.06)	0.81 (0.08)	0.80 (0.09)	0.84 (0.08)	36.0	<0.001	1 vs. 2 **, 1 vs. 3 **, 1 vs. 4 **, 2 vs. 4 *, and 3 vs. 4 *
Respiration rate (rep/min)	17.3 (3.29)	17.2 (4.54)	17.2 (5.28)	17.0 (4.77)	1.42	0.701	
AM (N)	0.97 (0.52)	0.84 (0.71)	0.87 (0.89)	0.99 (1.01)	3.87	0.276	
TM (N)	2.20 (1.09)	2.31 (0.92)	2.49 (1.66)	2.20 (1.40)	4.74	0.192	
AM + TM (N)	3.25 (1.55)	3.10 (1.45)	3.53 (2.30)	3.05 (2.05)	2.03	0.565	
AM/(AM + TM) (%)	29.8 (21.9)	24.7 (20.3)	28.5 (19.8)	32.3 (16.3)	8.18	0.042	
TM/(AM + TM) (%)	70.2 (21.9)	75.3 (20.3)	71.5 (19.8)	66.7 (16.3)	8.18	0.042	
Vital capacity (ml)	2612 (830)	2521 (700)	2344 (755)	2092 (741)	40.0	<0.001	1 vs. 4 **, 2 vs. 4 **, and 3 vs. 4 **
Vital capacity/height (mL)	16.3 (4.85)	15.7 (4.45)	14.6 (4.83)	13.2 (4.23)	43.5	<0.001	1 vs. 3 *, 1 vs. 4 **, 2 vs. 4 **, and 3 vs. 4 **
Vital capacity/weight (mL)	45.8 (13.7)	42.7 (16.3)	39.2 (18.3)	34.0 (11.6)	42.5	<0.001	1 vs. 4 **, 2 vs. 3 *, 2 vs. 4 **, and 3 vs. 4 *

Note: all values are presented as medians (interquartile ranges). Under “Between-group comparison”, 1 represents the 20–29 group, 2 represents the 30–39 group, 3 represents the 40–49 group, and 4 represents the 50–59 group. * *p* < 0.05, ** *p* < 0.01. AM, abdominal motion; TM, thoracic motion; AM + TM, the sum of abdominal motion and thoracic motion; AM/(AM + TM), the abdominal motion divided by the sum of abdominal and thoracic motions; BMI, the weight in kilograms divided by the square of the height in meters; N, Newtons; kg, kilograms; cm, centimeters; ml, milliliters. The ages shown are in years.

**Table 4 medicina-59-01024-t004:** Regression analysis for the prediction of the respiratory movements in women.

	Dependent Variables	Independent Variables	B	95% CI	β	*p* ^β^	R^2^	F	*p* ^F^
20–39 years old	AM/(AM + TM)	Age	−0.005	−0.009, 0.000	−0.167	0.037	0.046	4.664	0.011
		Height	−0.005	−0.008, 0.000	−0.191	0.018			
	TM/(AM + TM)	Age	0.005	0.000, 0.009	0.167	0.037	0.046	4.664	0.011
		Height	0.005	0.000, 0.008	0.191	0.018			
40–59 years old	AM/(AM + TM)	Age	0.003	0.001, 0.006	0.163	0.011	0.022	6.585	0.011
	TM/(AM + TM)	Age	−0.003	−0.006, −0.001	−0.163	0.011	0.022	6.585	0.011
20–59 years old	Vital capacity	Age	−21.08	−26.907, −15.250	−0.329	<0.001	0.174	41.65	<0.001
		Height	27.09	17.88, 37.93	0.254	<0.001			

Note: as the stepwise regression analysis was performed, the non-significant predictors were removed automatically. AM, abdominal motion; TM, thoracic motion; AM + TM, sum of the abdominal motion and thoracic motion; AM/(AM + TM), abdominal motion divided by the sum of the abdominal and thoracic motions; VC, vital capacity.

**Table 5 medicina-59-01024-t005:** Differences between the men’s and women’s respiratory movements.

Group	Respiratory Movements	Men	Women			
20–29 years		*n* = 28	*n* = 40	z	*p*	Effect size
	AM (N)	1.57 (1.70)	0.97 (0.52)	−1.670	0.095	0.20
	TM (N)	2.85 (1.55)	2.20 (1.09)	−3.265	0.001	0.40
	AM + TM (N)	4.39 (3.10)	3.25 (1.55)	−2.704	0.007	0.33
	AM/(AM + TM) (%)	28.3 (23.1)	29.8 (21.9)	−0.536	0.592	0.06
	TM/(AM + TM) (%)	71.7 (23.1)	70.2 (21.9)	−0.536	0.592	0.06
	Vital capacity (mL)	4169 (848)	2612 (830)	−6.704	0.000	0.81
	Vital capacity/height (mL)	23.5 (4.44)	16.3 (4.85)	−6.355	0.000	0.77
	Vital capacity/weight (mL)	51.2 (16.1)	45.8 (13.7)	−2.480	0.013	0.30
30–39 years		*n* = 42	*n* = 114	z	*p*	Effect size
	AM (N)	1.28 (1.21)	0.84 (0.71)	−2.861	0.004	0.23
	TM (N)	3.21 (1.70)	2.31 (0.92)	−3.352	0.001	0.27
	AM + TM (N)	4.32 (2.02)	3.10 (1.45)	−4.866	0.000	0.39
	AM/(AM + TM) (%)	31.1 (26.1)	24.7 (20.3)	−0.639	0.523	0.05
	TM/(AM + TM) (%)	68.9 (26.1)	75.3 (20.3)	−0.639	0.523	0.05
	Vital capacity (mL)	3833 (1207)	2521 (700)	−7.323	0.000	0.59
	Vital capacity/height (mL)	22.5 (6.66)	15.7 (4.45)	−6.454	0.000	0.52
	Vital capacity/weight (mL)	48.4 (18.1)	42.7 (16.3)	−1.931	0.053	0.16
40–49 years		*n* = 47	*n* = 146	z	*p*	Effect size
	AM (N)	2.02 (1.80)	0.87 (0.89)	−5.726	0.000	0.41
	TM (N)	3.34 (2.41)	2.49 (1.66)	−3.456	0.001	0.25
	AM + TM (N)	5.88 (4.01)	3.53 (2.30)	−5.302	0.000	0.38
	AM/(AM + TM) (%)	37.1 (21.8)	28.5 (19.8)	−3.648	0.000	0.26
	TM/(AM + TM) (%)	62.9 (21.8)	71.5 (19.8)	−3.648	0.000	0.26
	Vital capacity (mL)	3335 (944)	2344 (755)	−6.956	0.000	0.50
	Vital capacity/height (mL)	19.3 (5.58)	14.6 (4.83)	−5.991	0.000	0.43
	Vital capacity/weight (mL)	45.4 (20.9)	39.2 (18.3)	−2.475	0.013	0.18
50–59 years		*n* = 46	*n* = 101	z	*p*	Effect size
	AM (N)	2.37 (1.83)	0.99 (1.01)	−6.070	0.000	0.50
	TM (N)	3.46 (2.31)	2.20 (1.40)	−3.898	0.000	0.32
	AM + TM (N)	5.96 (2.34)	3.05 (2.05)	−5.686	0.000	0.47
	AM/(AM + TM) (%)	42.3 (20.7)	32.3 (16.3)	−4.307	0.000	0.36
	TM/(AM + TM) (%)	57.7 (20.7)	66.7 (16.3)	−4.307	0.000	0.36
	Vital capacity (mL)	3120 (677)	2092 (741)	−7.117	0.000	0.60
	Vital capacity/height (mL)	18.1 (3.70)	13.2 (4.23)	−6.258	0.000	0.53
	Vital capacity/weight (mL)	41.2 (12.8)	34.0 (11.6)	−3.003	0.003	0.25

Note: all values are presented as medians (interquartile ranges). AM, abdominal motion; TM, thoracic motion; AM + TM, sum of abdominal motion and thoracic motion; AM/(AM + TM), abdominal motion divided by the sum of the abdominal and thoracic motions; TM/(AM + TM), thoracic motion divided by the sum of the abdominal and thoracic motions; N, Newtons; ml, milliliters.

## Data Availability

The data that support the findings of this study are available on request from the corresponding author, Wenming Liang.

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
