# Peer review of "Respiratory Movements at Different Ages"

_medicina, 2023, doi:10.3390/medicina59061024_

Round 1
Reviewer 1 Report
Comments
Manuscript ID: medicina-2354730
Introduction:
Please consider improving the writing in the first paragraph to be concise. I think the type of exercise the author cited may not be relevant to this research at all.
Materials and Methods:
1. In the part of “Trial Design and Participants”, please consider improving the writing to be concise and explain more about the reasons for being excluded from the research. However, to make the protocol of this study easier to understand, please consider making this part a flowchart of the conduct of the research.
2. Inclusion and exclusion criteria do not need to be written in the order that should be described as an overview.
3. Please check Figure 1A which does not match the figure shown in the article as it was found to be “B”.
Results:
- Tables 1 and 3, please summarize the overall baseline characteristics of the participants before describing other results.
- Please check the consistency of Tables 2 and 4 whether the age information should be entered in both tables or not.
Limitation:
Is this research collection during the covid-19 outbreak? If yes, did the researcher collect volunteer data on this issue or not because post-covid patients are likely to affect breathing?
Author Response
Dear Reviewer,
Thank you very much for your time and effort in reviewing our manuscript, as well as for your valuable comments. We carefully revised our manuscript. Our responses to your comments, suggestions, and questions are as follows.
Introduction:
Your comments: Please consider improving the writing in the first paragraph to be concise. I think the type of exercise the author cited may not be relevant to this research at all.
Our response: We agree with you that the introduction was not concise, and the type of exercises was not very important. Therefore, we revised the introduction and removed the parts regarding the exercises.
Materials and Methods:
Your 1st comments: In the part of “Trial Design and Participants”, please consider improving the writing to be concise and explain more about the reasons for being excluded from the research. However, to make the protocol of this study easier to understand, please consider making this part a flowchart of the conduct of the research.
Our response: We agree with your suggestion and added a flowchart to present the sampling methods and exclude criteria.
Your 2nd comments: Inclusion and exclusion criteria do not need to be written in the order that should be described as an overview.
Our response: We removed the number of orders.
Your 3rd comment: Please check Figure 1A which does not match the figure shown in the article as it was found to be “B”.
Our response: Thank you for your attentiveness. We revised the figure.
Results:
Your 1st comment: Tables 1 and 3, please summarize the overall baseline characteristics of the participants before describing other results.
Our response: Thank you for your suggestion. We summarized the overall baseline characteristics of the participants for men and for women.
Your 2nd comment: Please check the consistency of Tables 2 and 4 whether the age information should be entered in both tables or not.
Our response: Table 2 presented the result from regression analysis for wen as one group from 20 to 59 years old percipients. Whereas Table 2 presented the results from women participants in two age groups that, one group from 20 to 39 years and another from 40 to 59 years. Therefore, the age information from Table 2 and Table 4 are different. The reason for that is clarified in the manuscript (see lines 323-326).
Limitation:
Your questions: Is this research collection during the covid-19 outbreak? If yes, did the researcher collect volunteer data on this issue or not because post-covid patients are likely to affect breathing?
Our responses: Yes, the data was collected during the COVID-19 outbreak between June to December 2021. Unfortunately, we did not ask participants if they had COVID-19. Therefore, we added one limitation: Thirdly, the data were collected from June to December 2021 during the COVID-19 outbreak. The World Health Organization reported that shortness of breath or difficulty breathing are common symptoms of post COVID-19 conditions. Unfortunately, we did not record whether participants had ever had coronavirus, but excluded people who continued to have cardiopulmonary diseases.
Best regards
All authors
Reviewer 2 Report
Some noun phrases lack a determiner before them. Consider adding an article.
The grammar needs to be checked.
Sentence punctuation needs adjustment
Improve the description of the sampling procedure
The grammar needs to be checked.
Sentence punctuation needs adjustment
Improve the description of the sampling procedure
Improve the problematization of the paper in the introduction
The English need minor adjustments, especially in the words: significant and not significant in the discussion. These words ask for the p-value to prove what is being stated.
Author Response
Dear Reviewer,
Thank you very much for your time and effort in reviewing our manuscript and for your valuable comments. We carefully revised our manuscript. Our responses to your comments, suggestions, and questions are as follows.
- Regarding grammatic errors.
Your comments: Some noun phrases lack a determiner before them; consider adding in an article; The grammar needs to be checked; sentence punctuation needs adjustment.
Our response: We agree with you and asked a professional language editing company to correct our grammatical errors.
- Improve the description of the sampling procedure
Our response: We detailed the sampling procedure and added a flowchart for a better description.
- Improve the problematization of the paper in the introduction
Our response: We have made a large number of changes in the introduction to improve the problematization and tried to make the introduction more concise.
- The correction you made in the DPF file.
Our response: We really appreciate your efforts in correcting our manuscript.
(1) We revised the manuscript according to your correction on punctuation, preposition, tense, and other grammatic errors.
(2) We rephrased the sentence as you suggested (on Lines 88-93 in the new version in the PDF file)
(3) We chose participants from 20-59 years of age because we planned to investigate adults, and there were very few participants who were aged more than 59 years old.
(4) We rephrased the first sentence in the discussion (on Lines 137-139 in the new version in the PDF file)
(5) We added p values to better present the results in the discussion.
Best regards
All authors
Round 2
Reviewer 1 Report
Dear authors,
Thank you very much for your attention and determination to make your work so much better. The improvements returned are quite noticeable and much better.
Best.